# The Effect of a Polymer-Stabilized Latex Cobinder on the Optical and Strength Properties of Pigment Coating Layers

**DOI:** 10.3390/polym13040568

**Published:** 2021-02-14

**Authors:** Zhenghui Shen, Araz Rajabi-Abhari, Kyudeok Oh, Sooyoung Lee, Jiachuan Chen, Ming He, Hak Lae Lee

**Affiliations:** 1Program in Environmental Materials Science, Department of Agriculture, Forestry and Bioresources, College of Agriculture and Life Sciences, Seoul National University, Seoul 08826, Korea; zhshen@snu.ac.kr (Z.S.); araz61@snu.ac.kr (A.R.-A.); 2Research Institute of Agriculture and Life Sciences, Seoul National University, Seoul 08826, Korea; ogd0310@snu.ac.kr; 3Songkang Industrial Co. Ltd., Eumseong-gun, Chungbuk-do 27651, Korea; leesooyoung4761@daum.net; 4State Key Laboratory of Biobased Material and Green Papermaking, Qilu University of Technology, Shandong Academy of Sciences, Jinan 250353, China; chenjc@qlu.edu.cn (J.C.); heming8916@qlu.edu.cn (M.H.)

**Keywords:** polymer-stabilized latex, pigment coating, cobinder, coating color, coating layer

## Abstract

Coated paper with a porous coating layer may have enhanced light-scattering ability and thus favorable optical properties. However, the increased porosity of such a coating layer is likely to decrease the strength of the coated paper, thereby adversely affecting the quality of the paper in the printing and converting processes. In this research, polymer-stabilized (PS) latex was prepared and used as a cobinder for the pigment coating of the paper. The PS latex particles were colloidally stabilized by a 3:1 mixture of starch and polyvinyl alcohol. The influence of the PS latex cobinder on the viscosity, sedimentation, and consolidation of coating colors was investigated. In addition, the effect of the cobinder on the properties of coating layers, namely, their porosity and surface, optical, and tensile properties, was examined. The results revealed that the PS latex cobinder formed microstructures in the coating colors and affected their viscosity. The addition of PS latex also led to enhanced interactions between coating color components, which affected the consolidation of the coating color, resulting in the formation of dried coating layers with greater porosity and improved optical properties (i.e., higher brightness and opacity) relative to coatings without the PS latex cobinder. Furthermore, the addition of PS latex improved the tensile strength of the coating layers, which was attributable to the small size and the polymeric protective shell of the cobinder particles. Thus, these results show that this PS latex cobinder has the potential to be used for the production of high-quality coated paper products.

## 1. Introduction

Pigment coating is used to improve the optical and printing properties of paper [1]. In pigment coating, a coating color (i.e., an aqueous mixture of pigments, binders, and other additives) is applied on the surface of a base paper to form a thin layer of coating composite after drying [2]. A pigment coating generally improves the smoothness, gloss, brightness, and opacity of a paper, thereby affording a higher product quality [3,4]. Such a coating also improves the printing properties of a paper by transforming a rough and non-uniform paper surface to a smooth and uniform surface.

The optical and printing properties of a coated paper are highly dependent on the void structure and strength of the coating layer [5,6]. In particular, the more porous a dried coating layer is, the more it scatters light, and thus, the higher the brightness and opacity of the paper are [5]. Thus, a coated paper must have a porous dried coating layer to exhibit good optical properties, mainly brightness and opacity. However, an increase in coating porosity negatively affects the mechanical strength of a coating layer [7]. This can lead to problems, as a coated paper must have sufficient strength, in the form of picking and folding resistance and dry- and wet-rub resistance, to withstand converting processes, such as printing and folding [8,9]. Therefore, there is an urgent need for a method to generate paper coatings that exhibit improved optical and strength properties simultaneously.

The structure of a coating layer is greatly influenced by the viscoelastic properties of the coating color involved. These properties are generally dependent on the type, shape, size, and relative proportions of coating pigments and binders and their interactions with other coating components [10,11,12,13,14,15,16,17]. Flocculation and the formation of microstructures in a coating color lead to the formation of a more porous coating layer, as they restrict the mobility of particles during drying [15]. Hence, the surface and optical properties of a coated paper can be optimized by the use of coating components that generate a coating color with a sufficient level of microstructures.

Synthetic polymer latexes are widely used as binders in paper coatings, as they provide good bonding strength and improve the runnability of coatings [4]. Surface-active agents are typically used to colloidally stabilize these latexes via electrostatic protection of latex particles. However, the use of these agents may lead to the generation of foams or the loss of stability under high shear rates and is environmentally unfriendly [18], which motivates coating technologists to develop new latexes.

Polymeric stabilizers can also be used to provide colloidal stability to synthetic latexes [19,20]. In previous studies, we prepared a polymer-stabilized (PS) latex additive stabilized using starch and polyvinyl alcohol (PVA) for the surface sizing of the paper [21,22]. The steric protection afforded by the polymers imparted superior colloidal stability to the latex across a wide pH range and also ensured a lower foaming tendency than conventional emulsion-polymerized latexes. Furthermore, the PS latex generated sized papers with improved mechanical properties, which was attributable to the use of starch and PVA stabilizers.

Recently, we developed a PS latex with a smaller average particle size of approximately 60 nm and utilized this latex as a cobinder in pigment coatings [23]. The protective shell of this PS latex consisted of a 3:1 mixture of starch and PVA, and it absorbed water in an alkaline environment, leading to increased water retention of coatings and substantially decreased foam generation by the coating colors. When styrene/butadiene (S/B) latex was partially substituted by PS latex, the foaming tendency of the coating color reduced. PS latex could generate less foam in the handling and coating process because its steric stabilization was provided by polymeric shells rather than surfactants. These positive results suggested that it would be meaningful to investigate the effect of PS latex on the properties of coating layers. We expected that the presence of polymeric stabilizers around the spherical latex particles might increase the interparticle interactions in coating colors, and thereby alter the viscoelastic properties of coating colors and the structure and properties of coating layers.

In this research, PS latex was utilized as a cobinder for pigment coating as a partial substitute for conventional S/B latex, and the effect of this PS latex cobinder on the rheology and colloidal interactions of coating colors was investigated. In addition, the structural characteristics, brightness, and opacity of dried coating layers that contained and did not contain this PS cobinder were evaluated and compared. Finally, the influence of this PS cobinder on the strength of coating layers was investigated and analyzed. It can be expected that the hydrophilic polymers used to stabilize the latex (i.e., starch and PVA) can enhance the interactions between the coating components and thus generate more microstructures in coating colors, leading to the formation of more porous coating layers. Meanwhile, starch and PVA can enhance the strength of coating layers due to their strengthening effects. Therefore, the optical and strength properties of coating layers can be improved simultaneously. As far as the authors know, although important, such a concept or work has not been reported by others. We believe that this work will evoke interests in both academic and industrial fields due to the advantages of easiness, effectiveness, and versatility in improving the performance of coated paper products.

## 2. Materials and Methods

### 2.1. Materials

Commercial-grade emulsion-polymerized S/B latex (SBL, Trinseo Korea Ltd., Ulsan, Korea) with a solid content of 50% was used as the main binder. The average particle size of S/B latex was 147 nm, and its pH was 7.3. Ground calcium carbonate (GCC, Setacarb HG, Omya Korea Inc., Seoul, Korea) in slurry form and kaolin clay (Hydrogloss® 90, KaMin LLC., Macon, GA, USA) were used as pigments. Sodium polyacrylate (molar mass: 5100 g/mol, Sigma-Aldrich Korea Ltd., Yongin, Korea) was used as a dispersing agent. Hydrochloric acid (HCl, 1N, lab grade) and sodium hydroxide (NaOH, 1 N, lab grade) (both from Samchun Chemical Co., Ltd., Seoul, Korea) were used to adjust the pH of coating colors. Deionized (DI) water was used in all experiments.

### 2.2. Polymerization

Styrene (S), butyl acrylate (BA), and acrylic acid (AA) (Sigma-Aldrich Korea Ltd., Yongin, Korea) were used as monomers for the polymerization of PS latex. The initiator for the polymerization process was 2,2′-azobis(2-amidinopropane) dihydrochloride (AAPH, Fujifilm Wako Pure Chemical Corp., Osaka, Japan). Polyvinyl alcohol (PVA, Poval, Kuraray Co., Ltd., Tokyo, Japan) with a hydrolysis degree of 98–99.5% and a molecular weight of 75,000–80,000 g/mol and oxidized starch (OS, Ingredion Korea Inc., Incheon, Korea) were used as stabilizers for the latex. Ammonium hydroxide was used as a buffer for pH regulation. The details of the polymerization process were described in our previous study [23]. Briefly, monomers, stabilizers, and initiators were added to a 3-L reactor via metered pumps. The starch/PVA stabilizer ratio was 3:1. The polymerization processes were completed after 5 h, at which time the mixtures were cooled to ambient temperature and the pH adjusted to 6.5.

### 2.3. Electron Microscopy

The morphology of PS latex was recorded using a LIBRA 120 transmission electron microscope (Carl Zeiss, Oberkochen, Germany) at an operating voltage of 120 kV. The latex was diluted to a concentration of 0.01% *w*/*v*, and its pH was adjusted to 7 by NaOH. The latex particles were deposited onto formvar film-coated grids and then stained with 2% uranyl acetate solution. Residual water was removed with filter paper, and the samples were dried at room temperature before obtaining the images.

The surfaces of the resulting coating layers were studied using a field-emission scanning electron microscope (AURIGA, Carl Zeiss, Oberkochen, Germany).

### 2.4. Fourier-Transform Infrared Spectroscopy (FTIR)

The presence of hydrophilic protective shells on the PS latex particles was confirmed using attenuated total reflection–Fourier transform infrared (FTIR) spectroscopy (Nicolet 6700 Spectrometer, Thermo Scientific, Waltham, MA, USA) at 600–4000 cm^−1^. PS latex samples were cast in an aluminum dish, and latex films were obtained after drying at 70 °C.

### 2.5. Preparation and Characterization of Coating Colors

#### 2.5.1. Coating Color Preparation

The formulation of the coating colors is presented in Table 1. The coating color was prepared by vigorously stirring a mixture of all of the coating ingredients. The clay was dispersed in DI water, and 0.3 pph of sodium polyacrylate was applied as a dispersing agent. The ground calcium carbonate (GCC) slurry was mixed with this clay dispersion, and S/B latex and the PS latex cobinder were added. The solid content of the prepared coating color was adjusted to 65% with DI water. NaOH was used to adjust the pH of the final coating color to 9.

#### 2.5.2. Viscosity Measurement

The low shear viscosity of the coating colors was evaluated at 100 rpm using a Brookfield viscometer (DV2T, Brookfield Engineering Laboratories, Inc., Middleboro, MA, USA), while the viscosity as a function of shear rate was measured using a stress-controlled rotational rheometer (CVO-100-901, Malvern Instruments Ltd., Worcestershire, UK) with cone-plate geometry (R = 40 mm, angle = 4°).

#### 2.5.3. Zeta Potential Measurement

The zeta potential of coating colors was evaluated using a Zetasizer (Nano-ZS, Malvern Instruments Ltd., Worcestershire, UK). To this end, the coating color medium was obtained by removing the sediment after centrifugation at 3000× *g* for 3 h with a large-capacity refrigerated centrifuge (Hanil Scientific Industrial, Daejeon, Korea). The coating medium was then diluted to 0.1% *w*/*w*, and samples were added to disposable folded capillary cells (DTS1060, Malvern Instruments Ltd., Worcestershire, UK) for the measurement.

#### 2.5.4. Sedimentation Testing

To measure the sediment volume of coating colors, 10 g of each sample was transferred into a 20 mL cylindrical plastic tube. Then, the tube was centrifuged at 3000× *g* for 3 h using a centrifuge (Hanil Scientific Industrial, Daejeon, Korea), and the supernatant was removed. The volume of the remaining sediment was then recorded.

#### 2.5.5. Drying Kinetics Analysis

Multispeckle diffusing-wave spectroscopy (MSDWS) is a reliable technique for studying the consolidation and drying kinetics of coating colors. The details of the measurement were explained in our previous work [15]. Briefly, the coating color was applied to a glass plate and then placed under the laser light of the MSDWS instrument (Horus, Formulaction Inc., Toulouse, France), followed by drying at a constant temperature of 23 °C and a relative humidity of 50%. The light scattered by the coating color was then detected with a camera. The interaction between coating color components, which is influenced by the presence of a PS latex cobinder, affects the Brownian motion of the particles in coating colors. The resulting differences in the dynamics and motion of the coating colors affect the intensity of the obtained MSDWS images.

### 2.6. Properties of Coating Layers

#### 2.6.1. Porosity Measurement

The porosity of the coating layers was evaluated by mercury intrusion porosimetry (AutoPore IV 9500, Micromeritics Instruments Corporation, Norcross, GA, USA). Each coating color was applied to a polyester film using an applicator bar with a gap size of 100 μm, and then dried at ambient temperature. As can be seen in Figure 1, the coating was uniform and the coated surfaces were free of defects. The thickness of the wet coatings was around 100 μm, and the thickness of the dried coating layer was approximately 20 μm. The porosity and pore size distribution of the dried coating layer were studied by measuring the volume of mercury that intruded into the pores of the dried coating layer. The pores were assumed to be cylindrical, with the diameter of this cylinder representing their size. The pore diameter was obtained from the external pressure data using Equation (1):(1)$$P=\frac{-4\gamma \cdot \cos\theta }{d}$$
where, *P* is the capillary pressure (Pa), *γ* is the mercury interfacial tension (N/m), *d* is the pore radius (m), and *θ* is the contact angle (°).

#### 2.6.2. Surface Property Measurement

The roughness of the coating layers was measured using an Lorentzen & Wettre (L&W) Parker Print-Surf (PPS) instrument (ABB, Kista, Sweden), while the brightness and opacity of the dried coating layers were determined with an L&W spectrophotometer (Elrepho, ABB, Kista, Sweden).

#### 2.6.3. Strength Property Measurement

Coating colors were applied to 30 μm thick polyester films using an automatic bar coater and were dried at room temperature for 12 h. The dried coating layers were then conditioned for 24 h at a constant temperature of 23 °C and a relative humidity of 50%. The resulting 50 μm thick coating layers were then carefully detached from the plastic films. The tensile strength of these samples was measured using a universal testing machine (Instron 5943, Instron Corp., Norwood, MA, USA). The width and length of the samples were 12 and 50 mm, respectively. At least 16 samples free of cracks and defects were prepared for tensile testing. The span length for the tensile testing was 30 mm, and the crosshead speed was 3 mm/min.

The effects of acrylate monomers of latex, starch, and PVA on the tensile properties of the coating layers were evaluated. For example, coating colors containing 9.8 parts of S/B latex and 1.2 parts of oxidized starch and/or PVA were prepared to show the effect of starch and PVA. All formulations of coating colors are shown in Table 2. Similarly, the coating colors had a pH of 9 and a solid content of 65%.

## 3. Results and Discussion

A PS latex consisting of a core polymer and a protective shell was prepared with a core-to-shell percentage ratio of 60:40. Butyl acrylate (60%), styrene (30%), and acrylic acid (10%) were used as monomers, and a 3:1 mixture of starch and PVA was used for stabilizing the synthesized latex. Then the PS latex was used as a cobinder in the coating formulation. Various amounts of S/B latex (main binder) in a coating color were replaced with PS latex, and the effect of these changes on the structure, surface properties, and mechanical properties of the coating layer was investigated systematically. The properties of the latexes are shown in Table 3. Compared to the sole addition of S/B latex (Figure 2a), weakly flocculated structures were formed upon the addition of this PS latex to the coating color (Figure 2b), which indicates that more interactions between coating color components had been established [23], and this will lead to a faster immobilization of the coating layer [15]. The morphology of PS latex is displayed by a transmission electron microscopy (TEM) image (Figure 2c).

More information about the properties of this latex cobinder, such as viscosity, particle size distribution, zeta potential as a function of pH, and viscosity at different pH values depending on the shear rate, is available in our previous work [23].

### 3.1. Viscosity and Interaction of Coating Suspensions

Figure 3a compares the Fourier transform infrared (FTIR) spectra of S/B latex and PS latex. In the FTIR spectrum of PS latex, the broad peak at 3700 to 3000 cm^−1^ was assigned to the stretching vibration of –OH groups, which indicated the presence of hydrophilic starch and PVA. This polymeric protective shell influenced the interactions between coating color components, and thus altered the viscoelastic properties of the coating color.

The low shear viscosity of the coating colors containing S/B latex as a sole binder, the formulation of which are shown in Table 1, was measured and compared with that of coating colors containing 1, 3, and 5 parts per hundred (pph) of the PS latex cobinder (Table 4). The low shear viscosity of the coating color increased as S/B latex was substituted with the PS latex cobinder, and the increment was larger with the increase in the substitution dosage of the PS cobinder. This was due to the presence of smaller particle-sized PS latex with a greater surface area, which led to more interactions with other coating ingredients [11]. In addition, the deprotonation of carboxylic acids in the core polymers of the PS latex resulted in the expansion of the core particle and increased the volume fraction of the dispersed phase [24]. Free polymer stabilizers that were not absorbed onto the PS latex cobinder likely affected the viscosity by inducing the depletion flocculation of the coating color [23,25].

Figure 3b shows the effect of substituting S/B latex with the PS cobinder on the viscosity of the resulting coating color as a function of shear rate. Notably, all of the resulting coating colors exhibited shear-thinning behavior: the more PS latex was present in the system, the higher was the viscosity of the coating colors. Besides, as the amount of PS latex was increased, the shear dependency of the coating colors increased, especially at low shear rates, indicating that the network formed in the coating color dispersed as the shear rate increased. Furthermore, the zeta potential of the coating colors was not significantly influenced by the addition of the PS latex cobinder, indicating that the components in the coating formulation were negatively charged and had similar zeta potential values [23], which suggested that the interaction between coating components was not due to electrostatic interactions (Figure 3c) [26,27]. Finally, it can be seen that the coating color that contained only S/B latex as a binder formed the most compact sediment (Figure 3d). When S/B latex was replaced with the PS latex cobinder, the sediment volume increased, indicating that a weakly flocculated network had formed due to the enhanced interactions between coating color components.

### 3.2. Drying Kinetics and Structure of Dried Coating Layer

Multispeckle diffusing-wave spectroscopy (MSDWS) was used to study the formation of structure in coating colors in the early stage of the immobilization of the wet coating components (Figure 4a) [15]. As can be seen, the Brownian motion of particles, as represented by the intensity difference between the speckled patterns, decreased continuously until each coating layer was entirely immobilized [28,29]. The coating color containing only S/B latex as a binder required 187 s for full immobilization, but the partial substitution of S/B latex with the PS latex cobinder accelerated the immobilization. This effect was most pronounced when 5 pph of S/B latex was replaced with the PS cobinder, as this resulted in a coating color that immobilized immediately after being applied to the glass substrate, indicating that this proportion of PS latex generated a coating color that formed the most extensive interactions between coating components. Notably, the speckle pattern of the coating color containing only S/B latex as a binder exhibited marked fluctuations, which indicated that there was extensive free movement of the particles within this coating color. However, the free movement of the particles in the coating color decreased with the increasing addition of PS latex.

The early immobilization of the wet coating layer is a key factor that influences the pore structure of the dried coating layer [30]. It has been proposed that early immobilization of the wet coating also hinders the migration of binders to the surface of the coating layer and promotes the formation of porous coatings with high surface roughness. To examine this possibility, the porosity of dried coating layers was evaluated by mercury intrusion porosimetry (Figure 4b). As can be seen, a more porous coating was formed when the PS latex cobinder was added to the coating color. As the proportion of the PS latex cobinder in the coating was increased, the porosity of the resulting coating increased due to a stronger interaction among coating components that stimulated the rapid immobilization of the wet coating. The cumulative pore volumes of coating layers and pore size distribution curves are shown in Figure 4c,d, respectively. The cumulative pore volume of the coating layer increased with the addition of the cobinder due to the structure-forming characteristics of the latter [14,15]. This agrees with the results in Figure 4b. Moreover, the greatest change in pore size due to the addition of the PS cobinder was observed in pore size diameters of 50 to 80 nm. This result also showed that most of the pores were 15–100 nm in diameter. It was believed that the higher porosity of the coating layer was caused by the microstructures in coating colors, and these microstructures or agglomerates would be effective in the scattering of light, thus improving the optical properties of the dried coating layers.

The influence of PS latex on the morphology of dried coating surfaces was evaluated by microscopic imaging. The samples were prepared by applying the coating color to a plastic film, and the resulting coated films were dried at room temperature. Field-emission scanning electron microscopy (FE-SEM) images of the dried coating layers are shown in Figure 5. As can be seen, the coating layer that contained only the S/B latex binder had a relatively smooth surface (Figure 5a). The differences between Figure 5a,b are not obvious since the dosage of PS latex was only 1 pph. However, more uneven surfaces (Figure 5c,d) could be seen when the dosage of the PS latex cobinder increased to 3 and 5 pph. When more S/B latex was substituted with the PS latex cobinder, more open coating structures were formed, with the amount of microstructures increasing as well. The fact that more open coating surfaces were formed as more small-particle PS latex was substituted for larger-particle S/B latex was contrary to the theory that small binder particles migrate to the surface more easily, thereby forming more closed coating surfaces [31,32,33,34]. This may be attributable to the PS latex cobinder causing rapid consolidation of the coating color by restricting the migration of binder particles to the coating surface.

The surface, optical, and printing properties of coated papers are influenced by the properties of the dried coating layer [2,5,35]. Figure 6a shows the Parker Print-Surf (PPS) roughness of the dried coating layers. The PPS roughness of the coating layer increased as the proportion of the PS latex cobinder in the coating increased, and this observation was in good agreement with the FE-SEM results (Figure 5). This showed that increases in the interaction among coating components increased the porosity and roughness of the coating layer.

Notably, the porosity and void structure of a dried coating affect the light-scattering capacity and optical properties of the resulting coating layers [5,13]. When S/B latex was substituted with the PS latex cobinder, the brightness and opacity of the resulting coating layers increased (Figure 6b,c). This was attributable to the addition of the PS latex cobinder increasing the porosity of the coating layers, and more microstructures favored the scattering of light.

### 3.3. Tensile Properties of Coating Layers

A coating layer must have sufficient surface and tensile strength to withstand the forces that are applied or generated during the printing and converting processes of paper [7,36,37]. Thus, the effect of the addition of the PS latex cobinder on the tensile properties of coating layers was examined. It was found that the tensile strength of the sample increased when S/B latex was partially replaced with the PS latex cobinder, indicating that this latex cobinder reinforced the strength of binding between pigments (Figure 7a). This was an unexpected result, because there is typically an inverse relationship between the porosity and mechanical strength of coating layers [2,7,8]. This result was due to the protective shell of the cobinder, which consisted of starch and PVA. It is well known that PVA, which has a strong binding force, increases the mechanical properties of coatings by enhancing their adhesive and film formation properties [38]. Starch can also increase the tensile strength of coating layers [39,40,41,42]. Moreover, the small particle size of the PS latex cobinder likely contributed to the increase in the bonding strength between pigments [43,44]. This result shows that the presence of this latex cobinder increased the strength of the coating layers and thus allowed a lower binder content to be used in coatings while also improving the porosity and optical properties of the coating layer.

The role of the components in the PS latex cobinder in improving tensile strength was examined. As the PS latex cobinder consisted of a styrene/acrylate (S/A) core and a stabilizing layer of PVA and starch, the effect of these three components on the tensile strength of the coating layer was investigated (Figure 7b). When conventional S/A latex was used as the sole binder, the tensile strength of the resulting coating layer was slightly increased to 3.4 MPa compared with that of a coating layer when S/B latex was used as the sole binder. When 1.2 pph of oxidized starch was used as a cobinder, the tensile strength of the coating layer increased from 3.4 to 5.2 MPa, while the use of a 3:1 mixture (1.2 pph) of starch and PVA increased the tensile strength to 5.35 MPa. This result suggests that the S/A core and the polymeric layer consisting of starch and PVA all increased the tensile strength of the coating layer.

## 4. Conclusions

In this study, PS latex was prepared and used as a cobinder in pigment coatings of paper. A protective shell comprising a 3:1 mixture of starch and PVA was used to sterically stabilize the latex particles. The viscosity and sedimentation results of the coating color suggested that microstructures formed in wet coating colors due to the cobinder increasing the interactions between coating components. These microstructures also influenced the immobilization of the coating color. A more porous coating layer was obtained when PS latex was used as a cobinder, and this coating layer exhibited increased light scattering, brightness, and opacity compared with less porous coating layers. Even though the porosity of the coating layer was increased when S/B latex was partially replaced with PS latex, the tensile strength of the resulting coating layer was increased. This was attributable to the presence of starch and PVA in the protective shell of PS latex, as well as the smaller particle size of PS latex. This study showed that coating layers that contained PS latex had greater porosity and strength than those that did not, thereby confirming that PS latex may be a useful cobinder for paper coatings. The influence of PS latex cobinder-containing coatings on the performance of coated paper in converting processes, such as printing and folding, must therefore be investigated in future works.

## Figures and Tables

**Figure 1 polymers-13-00568-f001:**
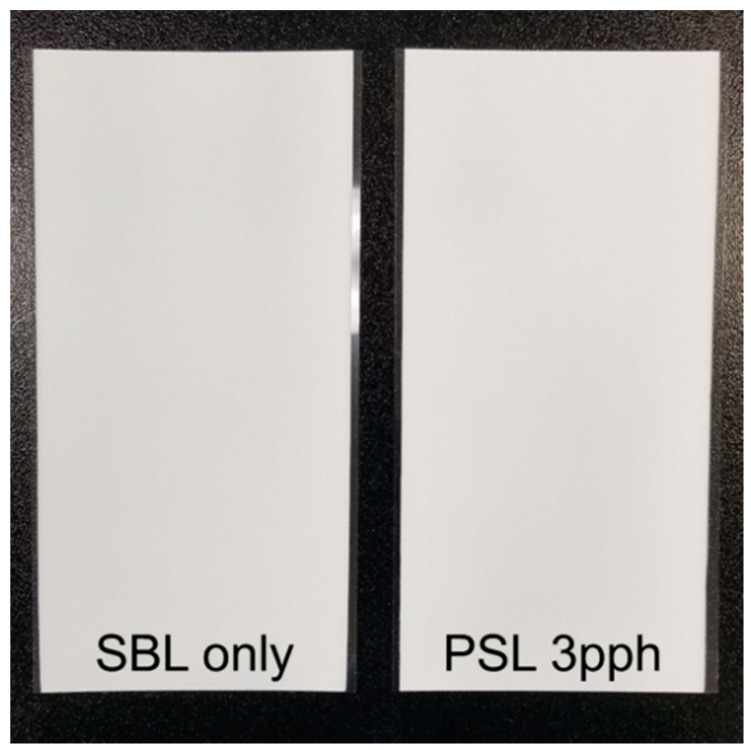
Digital image of coated polyester films: coatings without (**left**) and with 3 pph (**right**) the PS latex cobinder. SBL = styrene/butadiene (S/B) latex; PSL = polymer-stabilized (PS) latex.

**Figure 2 polymers-13-00568-f002:**
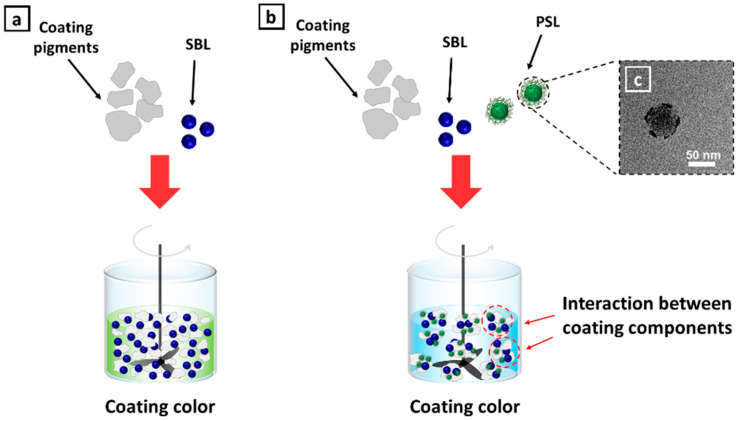
(**a**) Schematic illustrating the coating components and dispersion (i.e., coating color) obtained when only styrene/butadiene (S/B) latex (SBL) was used, (**b**) schematic illustrating the coating components and influence of the addition of the polymer-stabilized (PS) latex cobinder on a coating color, and (**c**) transmission electron microscopy image of a PS latex particle.

**Figure 3 polymers-13-00568-f003:**
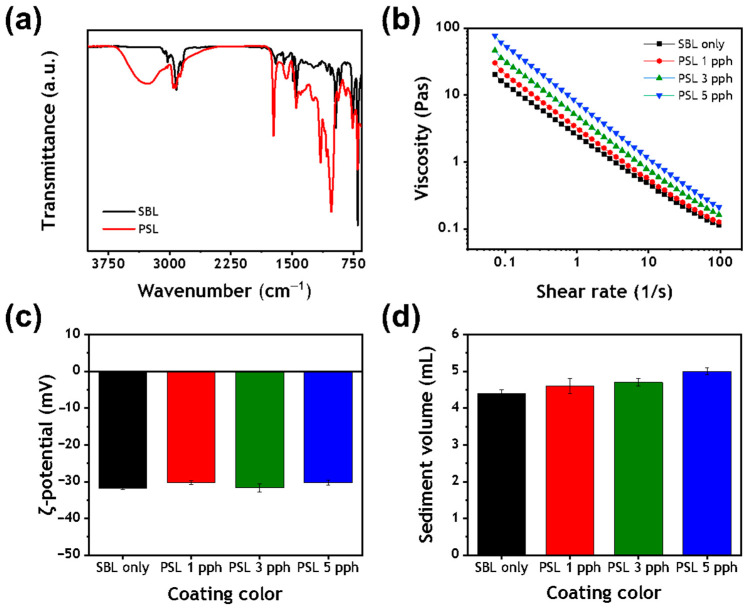
(**a**) Fourier transform infrared spectra of styrene/butadiene (S/B) and polymer-stabilized (PS) latex films, (**b**) the effect of PS latex dosage on the viscosity of coating color as a function of shear rate, (**c**) the zeta potential of coating colors, and (**d**) the sediment volume of coating colors.

**Figure 4 polymers-13-00568-f004:**
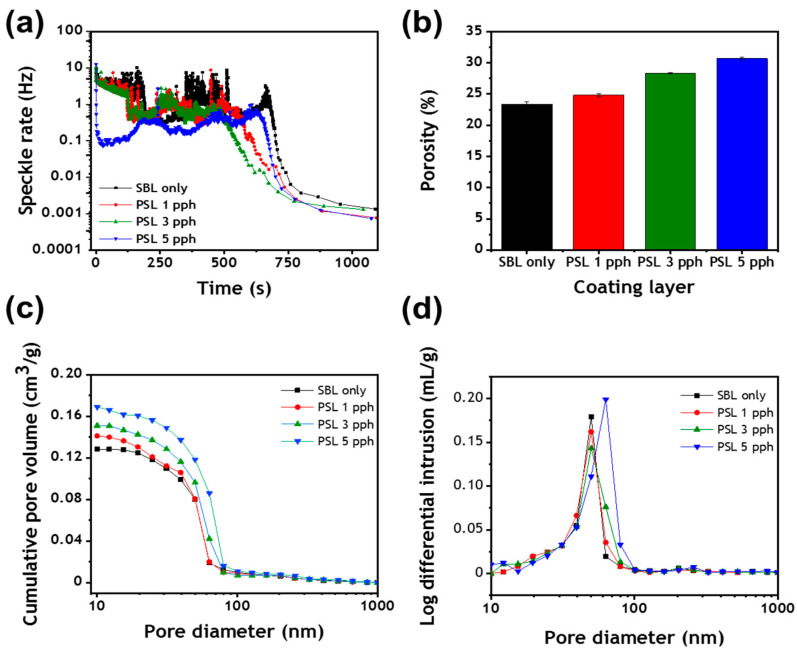
(**a**) Drying kinetics of coating layers, (**b**) porosity of coating layers, (**c**) cumulative pore volume of dried coating layers containing various proportions (parts per hundred; pph) of polymer-stabilized (PS) latex, and (**d**) effect of PS latex dosage on the pore size distribution of the coating layers.

**Figure 5 polymers-13-00568-f005:**
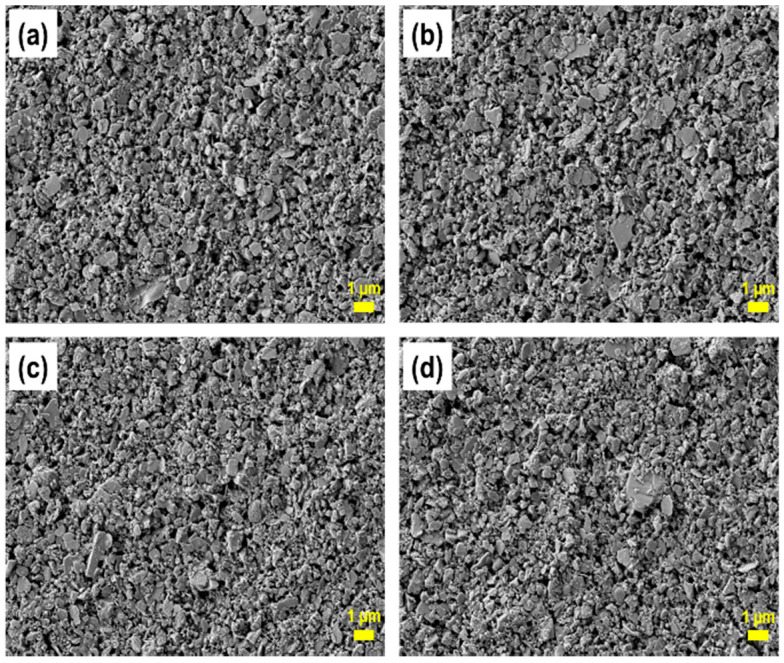
Field-emission scanning electron microscopy images of coating layers containing various amounts of PS latex: (**a**) 0 parts per hundred (pph), (**b**) 1 pph, (**c**) 3 pph, and (**d**) 5 pph.

**Figure 6 polymers-13-00568-f006:**
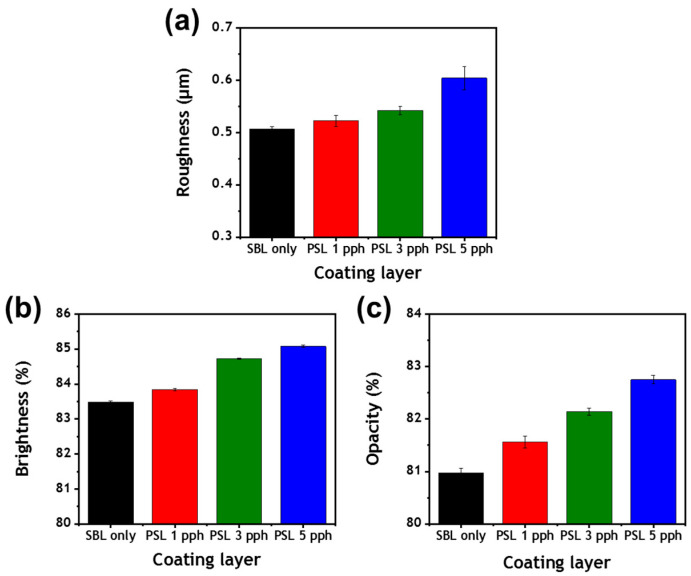
Surface and optical properties of coating layers containing various amounts of the PS latex cobinder: (**a**) roughness, (**b**) brightness, and (**c**) opacity of the coating layers.

**Figure 7 polymers-13-00568-f007:**
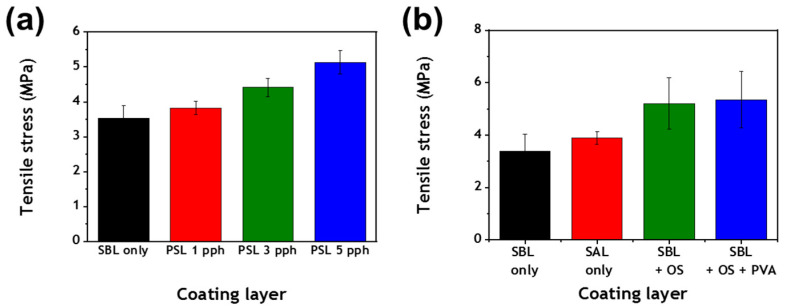
(**a**) Tensile stress of coating layers containing various amounts of the PS latex cobinder and (**b**) the effect of styrene/acrylate (S/A) latex (SAL), oxidized starch (OS), and polyvinyl alcohol (PVA) on the tensile stress of the coating layers.

**Table 1 polymers-13-00568-t001:** Formulation of coating colors.

Components		Amount (pph) ^1^
Pigment	GCC	80
	Clay	20
Binder	SBL	11/10/8/6
Cobinder	PSL	0/1/3/5

^1^ Component amounts are reported as parts per hundred (pph; by weight) based on 100 parts of pigment. GCC = ground calcium carbonate; SBL = styrene/butadiene (S/B) latex; PSL = polymer-stabilized (PS) latex.

**Table 2 polymers-13-00568-t002:** Formulations of coating colors for evaluating the effect of acrylate monomers of latex, starch, and PVA on the tensile strength of coating layers.

Coating Color		1	2	3	4
Components ^1^					
Pigment	GCC	80	80	80	80
	Clay	20	20	20	20
Binder	SBL	11	8	9.8	9.8
Cobinder	SAL	-	3	-	-
Additives	OS	-	-	1.2	0.9
	PVA	-	-	-	0.3

^1^ Component amounts are reported as parts per hundred (pph; by weight) based on 100 parts of pigment. SAL = styrene/acrylate (S/A) latex; OS = oxidized starch; PVA = polyvinyl alcohol.

**Table 3 polymers-13-00568-t003:** Properties of latexes.

Latex	Solids (%)	pH	Viscosity (cP)	Particle Size (nm)
SBL	50.0	7.3	364	147
PSL	48.0	6.5	201	60

**Table 4 polymers-13-00568-t004:** Effect of the PS latex cobinder on the viscosity and dewatering of coating colors.

Latex Dosage	SBL Only	PSL (pph) 1	PSL (pph) 3	PSL (pph) 5
Viscosity (cP)	210.0	246.0	414.0	550.8

## Data Availability

The data presented in this study are available on request from the corresponding author.

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
