# Peer review of "The Effect of a Polymer-Stabilized Latex Cobinder on the Optical and Strength Properties of Pigment Coating Layers"

_polymers, 2021, doi:10.3390/polym13040568_

Round 1

Reviewer 1 Report

The manuscript presents an interesting and systematic study of coatings that improve paper quality. In particular, a polymer-stabilized (PS) latex was prepared and used as a binder in pigment coatings on paper. A protective shell containing a mixture of starch and PVA in a 3:1 ratio was used for steric stabilization of latex particles. It was shown that coating layers that contained PS latex had greater porosity and strength than those that did not contain latex, thereby confirming that PS latex could be a useful binder for paper coatings. After careful reading, I think that the manuscript can be published in Polymers.

Author Response

Reviewer #1: The manuscript presents an interesting and systematic study of coatings that improve paper quality. In particular, a polymer-stabilized (PS) latex was prepared and used as a binder in pigment coatings on paper. A protective shell containing a mixture of starch and PVA in a 3:1 ratio was used for steric stabilization of latex particles. It was shown that coating layers that contained PS latex had greater porosity and strength than those that did not contain latex, thereby confirming that PS latex could be a useful binder for paper coatings. After careful reading, I think that the manuscript can be published in Polymers.

Response: Thanks for your comments. It is our honor to get your positive assessments. Thanks!

Reviewer 2 Report

Journal: Polymers
Manuscript ID: polymers-1099663
Title: The Effect of A Polymer-stabilized Latex Cobinder on the Optical and Strength Properties of Pigment Coating Layers
Comments: In this work, authors investigated viscosity, porosity, surface morphology, brightness, and tensile strength of polymer coating layers composed of coloring pigments, styrene/butadiene latex, and latex stabilized by starch and polyvinyl alcohol. By replacing latex particles with hydrophilic polymer-stabilized latex, it was observed that the porosity, brightness, and tensile strength were enhanced. Thorough experiments were performed to reveal the differences between the samples, compared to the control sample. This could be an interesting work for the communities of polymer composites and coatings. However, there are several issues to be resolved before publication of this manuscript in Polymers. See blow:
1/ Although a large set of work were described, it is not clear what significance of this manuscript is expected from the introduction. Significance and novelty of this work should be emphasized in the manuscript.
2/ Table 1: Solid content and pH can be presented in different columns, respectively.
3/ In this work, light scattering at the coating layer is a good effect. However, "scattering of light" may not be desirable in many applications of coating. I suggest that the definition of "optical properties" used in this work should be clearly stated in the manuscript. Use of different terms than "optical properties" would be more desirable.
4/ Line 292: Pore sizes of 15 and 100 nm in diameters would be less effective in scattering of the visible light with a wavelength of 400-700 nm. Instead, scattering by the particles and their agglomerates would be more effective in scattering. Please comment on this issue and revise the manuscript.
5/ Figure 4: I do not see any meaningful differences between the samples from the SEM images. Unless authors provide quantitative information on the difference from the SEM images, the claims of "roughness" and others (Line 295 and forward) should be revised.
6/ Figure 6: Stress-strain curves, in addition to the simple plots of tensile stresses of the samples, would be more meaningful and provide better idea in the manuscript.
7/ English correction must be conducted. There are some typos in the text. Some expressions and phrases are also awkward. Please read through the manuscript and correct errors by native speakers.

Author Response

Reviewer #2: In this work, authors investigated viscosity, porosity, surface morphology, brightness, and tensile strength of polymer coating layers composed of coloring pigments, styrene/butadiene latex, and latex stabilized by starch and polyvinyl alcohol. By replacing latex particles with hydrophilic polymer-stabilized latex, it was observed that the porosity, brightness, and tensile strength were enhanced. Thorough experiments were performed to reveal the differences between the samples, compared to the control sample. This could be an interesting work for the communities of polymer composites and coatings. However, there are several issues to be resolved before publication of this manuscript in Polymers. See blow:

1/ Although a large set of work were described, it is not clear what significance of this manuscript is expected from the introduction. Significance and novelty of this work should be emphasized in the manuscript.

Response: Agreed. It is well known that the properties of coating layers influence the performance of the coated paper and board. For instance, a more porous dried coating layer improves the light scattering, favoring the optical properties of coated products. However, an increase in coating porosity negatively affects the mechanical properties of a coating layer. Thus, the demand for improving both the strength and optical properties of coating layers turns to be a huge challenge.

This paper suggests the ability of a polymer-stabilized (PS) latex cobinder to increase the interactions between the components of coating colors. The hydrophilic polymers used to stabilize the latex, i.e. starch and polyvinyl alcohol (PVA) enhanced the interactions between the coating components and generated more microstructures in coating colors, leading to the formation of more porous coating layers. Importantly, starch and PVA enhanced the strength of coating layers due to their strengthening effects. Therefore, the optical properties and strength of coating layers were improved simultaneously, which might be valuable for improving the performance of coated paper and board. As far as the authors know, such a concept or work has not been reported by others.

Our work proved that it is effective to control the properties of coating layers by developing proper latex products and adopting proper coating formulations. We believe that this work will evoke interests in both academic and industrial fields, due to the advantages of easiness, effectiveness, and versatility in improving the performance of coated paper products.

The significance and novelty of this work have been emphasized in the revised manuscript, as shown in Line 98-107.

2/ Table 1: Solid content and pH can be presented in different columns, respectively.

Response: Thanks for your comment. The solid content and pH in Table 1 give some details about the preparation of coating colors. To improve the clarity and keep the uniformity of formats of Table 1 and Table 2, the information of solid content and pH in Table 1 has been moved to Line 149-151 in the revised manuscript. For the coating colors in Table 2, the information of solid content and pH have been added (Line 214-215).

3/ In this work, light scattering at the coating layer is a good effect. However, "scattering of light" may not be desirable in many applications of coating. I suggest that the definition of "optical properties" used in this work should be clearly stated in the manuscript. Use of different terms than "optical properties" would be more desirable.

Response: Agreed. In the pigment coating of paper, the optical properties of paper typically refer to brightness, opacity, etc. [1]. And the improved scattering of light generally favors these properties. To improve the understandability of the manuscript and avoid any misunderstanding, the definition of "optical properties" has been clearly stated in the revised manuscript, and “brightness, opacity, etc” have been used in necessary scenarios. Please see Line 28, 46, 48, 95-96 in the revised manuscript.

Reference:

[1] Jurič, I., Karlović, I., Tomić, I., & Novaković, D. (2013). PRINTING: Optical paper properties and their influence on colour reproduction and perceived print quality. Nordic Pulp & Paper Research Journal, 28(2), 264-273.

4/ Line 292: Pore sizes of 15 and 100 nm in diameters would be less effective in scattering of the visible light with a wavelength of 400-700 nm. Instead, scattering by the particles and their agglomerates would be more effective in scattering. Please comment on this issue and revise the manuscript.

Response: Agreed. The relevant expression and analysis have been updated in the revised manuscript, according to this comment. Please see Line 322-324 in the revised manuscript.

5/ Figure 4: I do not see any meaningful differences between the samples from the SEM images. Unless authors provide quantitative information on the difference from the SEM images, the claims of "roughness" and others (Line 295 and forward) should be revised.

Response: Thanks for the comment. The authors agree that it is not easy to see the obvious differences between the samples from the SEM images, especially between Figure 4a and b. However, more uneven surfaces (Figure 4c and d) can be seen when the dosage of PS latex cobinder increased to 3 pph and 5 pph, respectively. This was because the increased addition of PS latex generated more microstructures in coating colors. Moreover, the roughness results in Figure 5a confirmed the observation from SEM images. Despite this, the author revised the claims of "roughness" and others to improve the manuscript, as suggested by the reviewer. Please see these changes in Line 331-337.

6/ Figure 6: Stress-strain curves, in addition to the simple plots of tensile stresses of the samples, would be more meaningful and provide better idea in the manuscript.

Response: Thanks for the comment. The authors agree that the stress-strain curves can provide more information on the mechanical properties of materials. Nevertheless, the key of this research is to investigate how this polymer-stabilized latex influenced the properties of coating colors and thus the properties of coating layers. Our results indicated that the addition of this latex cobinder led to more interactions between the coating components, generating more microstructures in the coating color, which consequently improved the optical properties of coating layers. Besides, the hydrophilic polymers used to stabilize this latex cobinder, i.e. starch and polyvinyl alcohol (PVA) enhanced the strength of coating layers. Therefore, both the optical and strength properties of coating layers have been improved simultaneously. From this perspective, the tensile stress result is sufficient to achieve our research aim at this moment. However, we will put more interpretations into the mechanical properties of coated samples, e.g. stress-strain curves, in our future work.

7/ English correction must be conducted. There are some typos in the text. Some expressions and phrases are also awkward. Please read through the manuscript and correct errors by native speakers.

Response: Thanks for the comment. In fact, this manuscript had been polished by a professional English editing service before its submission to this journal. Probably some typos and improper expressions were made when we tried to improve the manuscript further. According to the review’s comment, the manuscript has been carefully checked to correct the typos and awkward expressions. Please see Line 25, 26, 30, 31, 41, 50, 56-58, 72-73, 92, 119, 133, 163, 190, 207, 225, 228, 305, 309, 364, 371, 374-375, 398, 403-404 in the revised manuscript.

Reviewer 3 Report

The manuscript by Phin et al. presents an interesting study on the effect of PS latex co-binder on the optical and mechanical properties of pigment coatings. The new PS latex has contributed to a more porous coating layer as well as improved tensile strength. The authors used various characterization techniques for performance assessment. The results shed light on potential applications for printing and paper coating. The experimental results and interpretations are sound. And thus I recommend publishing this manuscript. The specific comments are as follows:

1) Page 4, Line 171. The thickness of dry films was 20um. How was this measured? What is the thickness of the wet coatings?

2) I suggest authors to include a picture of paper coatings (printed films) from the formulation with the novel co-binder.

3) I recommend authors to include a discussion on the coating stability over time/cycles.

4) The author mentioned the PS latex in this study has the core-shell morphology. However, the exact structure was not very clear. According to literature (Loxley, Vincent, J. Colloid Interface Sci. 1998, 208, 49), there are 4 possible two-phase particle morphology, namely (1) core/shell, (2) occluded, (3) "acorn" or "mushroom" like, (4) heteroaggregate. The mechanism of formation was attributed to the interplay of interfacial tensions between pairs of the three phases. Could authors help figure out which of the 4 type will best describe the latex morphology in this study? And also what is the dominating factor - the surface tension (which can be triggered by pH) or the elasticity of the shell?

5) Page 6, Line 223. I suggest authors to include more detailed peak assignment such as "which indicated the O-H stretching vibration of hydrophilic starch and PVA".

6) I suggest authors to include pH-dependent data for viscosity, average particle size and distribution of the PS latex.

7) What is the foaming tendencies of the coating using PS latex? I recommend authors to include a discussion in the main text.

Author Response

Reviewer #3: The manuscript by Phin et al. presents an interesting study on the effect of PS latex co-binder on the optical and mechanical properties of pigment coatings. The new PS latex has contributed to a more porous coating layer as well as improved tensile strength. The authors used various characterization techniques for performance assessment. The results shed light on potential applications for printing and paper coating. The experimental results and interpretations are sound. And thus I recommend publishing this manuscript. The specific comments are as follows:

1) Page 4, Line 171. The thickness of dry films was 20 um. How was this measured? What is the thickness of the wet coatings?

Response: Thanks for the comment. The coating layers were prepared by applying coating colors on the surface of polyester films using an applicator bar with a gap size of 100 μm and then dried at ambient temperature. Thus, the thickness of the wet coatings was around 100 μm. After drying, the coating layers were carefully detached from the surface of the polyester films. Then an L & W micrometer (ABB, Sweden) was used to measure the thickness of the prepared coating layers. Necessary details have been added in the revised manuscript, please see Line 185-187.

2) I suggest authors to include a picture of paper coatings (printed films) from the formulation with the novel co-binder.

Response: Agreed. A picture of paper coatings (printed films) from the formulation without/with this PS latex cobinder has been included in the revised manuscript (Line 195-198). The image is shown below. As can be seen, the coating was uniform and coated surfaces were free of defects. Then the coating layers were separated carefully and used for the measurements.

Figure 1. Digital image of coated polyester films: coatings without (left) and with 3pph (right) PS latex cobinder.

3) I recommend authors to include a discussion on the coating stability over time/cycles.

Response: Thanks for the comment. Intensive characterizations on the properties of this polymer-stabilized latex and the coating color containing this latex cobinder have been reported in our previous study [1] and the present study. However, it is a very important issue to test the stability of the coating colors over time or cycles. This is because a substantial portion of the coating color is generally recirculated after doctored off by the blade. It is an interesting topic to study the coating stability over time/cycles. At this moment, the authors did not notice any significant changes in viscosity of the coating colors containing PSL cobinder in our experiments, most probably because the addition rate was low. In fact, we have used this PSL cobinder for mill trials and did not see any troubles in production and rheology, either. This topic will be discussed in our next article which will include the results of mill trials. However, we agree that the long time stability of the coating colors should be tested more thoroughly, especially when the material tends to interact with other components.

References:

[1] Rajabi-Abhari, A., Shen, Z., Oh, K., Im, W., Kwon, S., Lee, S., & Lee, H. L. (2020). Development and Application of Nanosized Polymer-Stabilized Cobinders and Their Effect on the Viscoelastic Properties and Foaming Tendencies of Coating Colors. ACS omega, 5(16), 9291-9300.

4) The author mentioned the PS latex in this study has the core-shell morphology. However, the exact structure was not very clear. According to literature (Loxley, Vincent, J. Colloid Interface Sci. 1998, 208, 49), there are 4 possible two-phase particle morphology, namely (1) core/shell, (2) occluded, (3) "acorn" or "mushroom" like, (4) heteroaggregate. The mechanism of formation was attributed to the interplay of interfacial tensions between pairs of the three phases. Could authors help figure out which of the 4 type will best describe the latex morphology in this study? And also what is the dominating factor - the surface tension (which can be triggered by pH) or the elasticity of the shell?

Response: Thanks for the comment. We prepared the core/shell structure latex by using different monomer combinations for these two layers. We designed to have the (1) core/shell structure. You may see the references shown below. It is unfortunate, however, we are not allowed to describe any details of the polymerization process. And we do not have the answer to the final question yet.

References:

[1] Rajabi-Abhari, A., Shen, Z., Oh, K., Im, W., Kwon, S., Lee, S., & Lee, H. L. (2020). Development and Application of Nanosized Polymer-Stabilized Cobinders and Their Effect on the Viscoelastic Properties and Foaming Tendencies of Coating Colors. ACS omega, 5(16), 9291-9300.

[2] Abhari, A. R., Lee, H. L., Oh, K., Im, W., Lee, J. H., & Lee, S. (2018). Development and evaluation of suspension polymerised latex additive for surface sizing of paper. Appita: Technology, Innovation, Manufacturing, Environment, 71(4), 330.

5) Page 6, Line 223. I suggest authors to include more detailed peak assignment such as "which indicated the O-H stretching vibration of hydrophilic starch and PVA".

Response: Agreed. In the FTIR spectrum of PS latex, the broad peak at 3700 to 3000 cm−1 is assigned to the stretching vibration of –OH groups of the hydrophilic starch and PVA. It has been clarified in the revised manuscript. Please see the changes in Line 247-251.

6) I suggest authors to include pH-dependent data for viscosity, average particle size and distribution of the PS latex.

Response: Thanks for the comment. The suggested characterizations of this PS latex have been reported in our previous work [1], i.e. the viscosity, particle size and its distribution of PS-2 latex depending on pH. Moreover, the zeta potential as a function of pH and the viscosity at different pH values as a function of the shear rate are also available in that work. Please see more details in the followed reference. This has been mentioned in the revised manuscript (Line 235-237), thus the interested readers can refer to our previous work.

Reference:

[1] Rajabi-Abhari, A., Shen, Z., Oh, K., Im, W., Kwon, S., Lee, S., & Lee, H. L. (2020). Development and Application of Nanosized Polymer-Stabilized Cobinders and Their Effect on the Viscoelastic Properties and Foaming Tendencies of Coating Colors. ACS omega, 5(16), 9291-9300.

7) What is the foaming tendencies of the coating using PS latex? I recommend authors to include a discussion in the main text.

Response: Thanks for the comment. The foaming tendency of coating color containing this PS latex cobinder has been reported in our previous work [1]. The results showed that when S/B latex was partially substituted by this PS latex, the foaming tendency of the coating color reduced. PS latex could generate less foam in the handling and coating process because their steric stabilization was provided by polymeric shells rather than surfactants. The above discussion has been included in the revised manuscript (Line 83-86).

Reference:

[1] Rajabi-Abhari, A., Shen, Z., Oh, K., Im, W., Kwon, S., Lee, S., & Lee, H. L. (2020). Development and Application of Nanosized Polymer-Stabilized Cobinders and Their Effect on the Viscoelastic Properties and Foaming Tendencies of Coating Colors. ACS omega, 5(16), 9291-9300.